# Automatic Recognition of Multiple Emotional Classes from EEG Signals through the Use of Graph Theory and Convolutional Neural Networks

**DOI:** 10.3390/s24185883

**Published:** 2024-09-10

**Authors:** Fatemeh Mohajelin, Sobhan Sheykhivand, Abbas Shabani, Morad Danishvar, Sebelan Danishvar, Lida Zare Lahijan

**Affiliations:** 1Psychology Department, University of Aston, Birmingham B4 7ET, UK; 2Department of Biomedical Engineering, University of Bonab, Bonab 55517-61167, Iran; s.sheykhivand@tabrizu.ac.ir; 3Sports Science Department, Qom Branch, Islamic Azad University, Qom 37491-13191, Iran; 4College of Engineering, Design and Physical Sciences, Brunel University London, Uxbridge UB8 3PH, UK; 5Biomedical Engineering Department, Faculty of Electrical and Computer Engineering, University of Tabriz, Tabriz 51666-16471, Iran

**Keywords:** BCI, CNN, EEG, emotion, graph

## Abstract

Emotion is a complex state caused by the functioning of the human brain in relation to various events, for which there is no scientific definition. Emotion recognition is traditionally conducted by psychologists and experts based on facial expressions—the traditional way to recognize something limited and is associated with errors. This study presents a new automatic method using electroencephalogram (EEG) signals based on combining graph theory with convolutional networks for emotion recognition. In the proposed model, firstly, a comprehensive database based on musical stimuli is provided to induce two and three emotional classes, including positive, negative, and neutral emotions. Generative adversarial networks (GANs) are used to supplement the recorded data, which are then input into the suggested deep network for feature extraction and classification. The suggested deep network can extract the dynamic information from the EEG data in an optimal manner and has 4 GConv layers. The accuracy of the categorization for two classes and three classes, respectively, is 99% and 98%, according to the suggested strategy. The suggested model has been compared with recent research and algorithms and has provided promising results. The proposed method can be used to complete the brain-computer-interface (BCI) systems puzzle.

## 1. Introduction

Emotions can be thought of as a complex mental state that influences human physical behaviors and physiological processes [1]. Emotion recognition has the potential to be very useful in the field of BCI [2]. This issue bridges the gap between humans and intelligent devices, allowing them to monitor changes in human emotions and mitigate their negative effects on mental health [3]. Furthermore, emotions have an impact on both people’s health and their decision-making. Harmful emotions have serious consequences for people’s psychological and physiological health, including an increased risk of developing diseases like schizophrenia and depression [4,5]. Recent research has found that excessive exposure to negative emotions increases the likelihood of psychotic symptoms associated with schizophrenia. Furthermore, depression is associated with persistent increases in stress hormones, which are released during prolonged periods of negative emotions [5].

Scientists have employed diverse techniques to identify emotions, encompassing non-physiological factors such as speech and facial expressions. However, these variables are influenced by external factors and are unreliable [6]. Categorizing emotions for various applications, including BCI, has historically been a difficult task for psychological sciences that necessitates specialized knowledge. Emotions are typically divided into two categories: discrete models and dimensional models. The discrete model recognizes six different emotional states, including anger, disgust, fear, happiness, sadness, and surprise, all of which are represented by facial expressions. Additionally, the dimensional model includes valence, arousal, and neutral dimensions. Arousal represents the degree of emotional activation, whereas valence represents the positive or negative emotion. This hypothesis systematically describes emotions and is commonly used as background knowledge in numerous studies. Indeed, there exist multiple theories concerning the categorization of emotions [7,8]. For example, Ekman proposed six basic emotions: fear, anger, joy, sadness, disgust, and surprise, which he later expanded to include embarrassment, pride, excitement, shame, contempt, satisfaction, and amusement [9]. Russell also proposed sixteen emotions, which he mapped onto a two-dimensional plane of arousal (from pleasant to unpleasant) and valence (from calm to excited). Figure 1 illustrates Russell’s theory of emotion segmentation [10].

Music has been used to elicit emotions since ancient times. However, it is difficult to predict what emotions music evokes in a person. Furthermore, there are numerous methods for eliciting emotions, including words, images, sounds, and videos. However, music has a profound effect on people’s emotional states and is recognized as an exceptional tool for evoking emotions while also modulating neurophysiological processes. Compared to other stimuli, music has the ability to elicit deeper and more stable emotional responses in people. As a result, the stimulation in this study is primarily musical [11].

Emotions can be recognized in two ways: non-physiological and physiological. Non-physiological signals include tone of voice, body posture, facial movement, and others of a similar nature. The aforementioned signals can be mentally controlled and hidden, so they are associated with classification errors. In contrast, physiological signals include EEG, temperature, electrocardiogram (ECG), electromyogram (EMG), galvanic skin response (GSR), and respiration [12]. EEG is a non-invasive physiological signal that directly measures the electrical activity of the brain during various emotional states [12]. These signals have several advantages over other physiological signals. EEG has several advantages, including better time resolution, faster data collection and transfer, availability, and low registration costs. EEG signals can accurately measure the spontaneous signals produced by the human brain, which include various types of emotions [12]. Despite their advantages, EEG signals have limitations. These signals have low signal-to-noise ratios and limited spatial resolution. Several methods have been proposed for removing EEG signal artifacts. Some of these methods include repeating experiments to increase SNR [12]. Others use new methods, such as InvBase, to remove the baseline power prior to extracting the features [13]. On this basis, distinguishing emotions from EEG signals is a difficult and time-consuming task. This is why, in recent years, many models have been developed to automatically recognize emotions. These techniques rely on machine-learning (ML) algorithms, which are widely regarded as the primary approaches to expanding emotion-recognition methods based on EEG signals. These techniques involve extracting time, frequency, time-frequency, and statistical features like mean, variance, skewness, kurtosis, and nonlinear features from EEG signals. Prior research has employed different classifiers, including k-nearest neighbor (KNN), Adaboost, and RusBoost, to classify the selected feature vector. Some studies also employ deep learning, a technique that combines feature selection/extraction and classification simultaneously [13].

Understanding the relationship between EEG components and different emotions is a crucial scientific inquiry in cognitive neuroscience. It is essential for developing accurate recognition models. According to a report, there is a positive correlation between joyful music and the energy in the theta band (4∼7 Hz) near the midline of the prefrontal cortex. Furthermore, previous studies have found a correlation between the valence and arousal of musical stimuli and the asymmetry of frontal alpha (8∼13 Hz) activity. When subjects are exposed to happy music, the EEG activity in the left frontal region of the brain is more pronounced compared to the activity in the right frontal region. Conversely, playing sad music yields the opposite outcome. However, certain studies did not observe any effects in the alpha frequency range but instead discovered a correlation in the beta-2 frequency range (18∼22 Hz). The impact of high-frequency EEG, specifically the beta-2 band (18∼22 Hz), beta-3 band (22∼30 Hz), and gamma band (30∼45 Hz), on emotions has also been confirmed. The sensation of happiness leads to a reduction in beta-2 power in the frontal central areas, as well as a decrease in beta-3 and gamma power across the entire cortex. When experiencing anger, there is an increase in power in the beta-2, beta-3, and gamma bands in the frontal regions of both hemispheres [14].

Continuing the recent research in the field of automatic emotion recognition, each individual aspect is thoroughly examined, including their respective advantages and disadvantages. Sheykhivand et al. [14] proposed a novel intelligent method for identifying emotions from EEG signals. These researchers proposed a database based on musical stimulation and were successful in extracting three positive, negative, and neutral emotions from it. Their proposed deep network combined long short-term memory (LSTM) and convolutional neural networks (CNNs). In their proposed architecture, there were 10 convolutional layers and 3 LSTM layers, with point classification completed using two fully connected (FC) layers. These researchers achieved 97% accuracy using 12 channels of EEG signals. One advantage of this research was the proposed model’s usability in real-time environments. Baradaran et al. [15] proposed a new EEG-based model for recognizing emotions. These researchers proposed a model that used convolutional neural networks to distinguish between three positive, negative, and neutral emotions. The final accuracy reported in this study was approximately 98%. One advantage of this research is the low computational complexity, which increased classification speed. Baradaran et al. [16] proposed a new deep model that recognizes three different emotional states by combining type 2 fuzzy networks and deep convolutional networks. The use of fuzzy networks in conjunction with deep-learning networks made the proposed model highly resistant to environmental noise. In addition, a GAN was used in this study to augment the quantity of data before the data entered the deep network. The final accuracy reported in this study was 98%. Yang et al. [17] used Deap and Seed databases to automatically recognize emotions. Among the pre-processing steps taken in this study was the reduction of electrodes used in recording and extracting spectral features. In this study, the optimal features from the recorded signals were selected using two deep networks known as STCN and SITCN. In addition, the AM-softmax classifier was used to classify the feature vector. The final accuracy reported in this study was around 95%. Hussain et al. [18] investigated the automatic recognition of emotions from EEG signals. In their investigation, these scholars recorded EEG signals from 32 participants. Data preprocessing techniques used in this study included data augmentation and Z- normalization. The network used in this study was the LP-Vandy-CNN, which could automatically extract optimal features from EEG signals. The network’s classifier was also based on softmax. The accuracy achieved in this study is reported to be around 98%. Khubani et al. [19] used Deap and Seed-AV databases to automatically recognize emotions. The databases used in this study included 32 and 15 participants, respectively. These researchers pre-processed the data using discrete wavelet transform (DWT), statistical, and spectral analyses. The deep network proposed in this study could distinguish between different emotions with 97% accuracy. Peng et al. [20] used the Deap database to identify emotions from EEG signals. The database used in this study included 32 participants. After obtaining the EEG signals, pre-processing such as signal conversion from 1D to 2D and principal component analysis were performed on the recorded signals. These researchers proposed a model for feature selection and classification using temporal relative (TR) encoding. In this study, scores were also classified using the softmax function. The final accuracy reported by these researchers was approximately 95%. Xu et al. [21] introduced a novel graph theory-based model for emotion recognition. These researchers examined the relationship between music and brain networks using EEG signals from 29 participants under various music conditions. One of the research’s benefits is that it clarifies the relationship between music and the human brain, while one of its main drawbacks is its low classification accuracy. Alotaibi et al. [22] used EEG signals to determine emotion. These researchers conducted their research on the Deap and Seed databases. One of the databases mentioned in this study was pre-processed using the short-time Fourier transform (STFT). The proposed model used pre-trained Google Net networks. Additionally, emotions were classified using the nearest-neighbor method. The researchers eventually reached 96% accuracy. Qiao et al. [23] proposed a novel musical-stimulation-based model for classifying emotion states. These investigators used a prepared database to extract differential entropy characteristics. Then, using his proposed network, CNN-SA-BiLSTM, feature selection/extraction and classification were automated. These researchers reported an accuracy of approximately 93% for classifying four different emotional classes.

As is well known, numerous studies have been conducted in recent years to automatically recognize emotions. However, there are still some limitations to overcome. One of the initial limitations is the absence of an extensive EEG-signal-based musical stimulus database. Furthermore, the networks used in previous studies all have a high computational load and are unusable, despite their high classification accuracy. Moreover, many algorithms employ classical methods for feature selection/extraction and classification, which necessitate specialized knowledge of the subject/problem despite producing the desired results. This study attempts to address the aforementioned challenges by creating a standard database based on musical stimuli that uses a deep optimal network to accurately distinguish between positive, negative, and neutral emotions. The contribution of this article is summarized as follows:Presenting a deep automatic model to classify three different categories of positive, negative, and neutral emotions.Creating a database of EEG signals based on musical stimuli.The proposed deep model exhibits high resistance to environmental noise.Achieving the highest classification accuracy between positive, negative, and neutral emotions in comparison to other recent studies.Achieving the highest accuracy only based on the evaluation of 3 C_3_, C_4_, and P_z_ EEG channels makes it possible to use the proposed model in real-time environments.

The remaining sections of this research are arranged as follows: The second part looks at graph theory’s mathematical foundation. The third section is about the proposed method, which includes how to obtain the database, the pre-processing steps, the proposed architecture, and so on. The fourth section looks at the simulation results. Finally, the fifth section refers to the conclusion.

## 2. Background

The mathematical foundation of the employed algorithms will be examined in this part.

### 2.1. Brief of GANs

GANs have garnered significant attention from researchers in recent years. The presentation of these networks took place in 2014 by J. Goodfellow and his colleagues [24]. Generally, these networks primarily perform the role of unsupervised learning in the field of machine learning. The purpose of these networks is to produce data, which is achieved through two sub-networks: a generator (*G*) and a discriminator (*D*). Within the GAN, the generator and discriminator sub-networks engage in a competitive relationship. The generator attempts to produce synthetic data, while the discriminator endeavors to identify the synthetic data generated by the generator. To ensure the discriminator cannot identify its artificial nature, the data generated by the generator must be produced accordingly. The generator produces data by utilizing a random noise vector of a predetermined length. During the training phase, the aforementioned networks often seek to minimize the following relationship:(1)$$\log(1-D(G_{\left(Z\right)}))\;\begin{array}{ll} \frac{\min\max}{GD}V(G,D)= & E_{x}-P_{data}[\log D(x)]\\ & +E_{pz(z)}[\log(1-D(G(Z))] \end{array}$$

To solve the problem, the discriminator must be obtained in a way that allows it to distinguish between real and fake data. This equation is not solvable analytically and necessitates iterative algorithms for its solution. Additionally, to prevent overfitting the data, the optimization of function *D* will be accompanied by the optimization of function *G* for each *k* [24].

### 2.2. Brief of the Combination of Graph Theory and Deep Convolutional Networks

Machine learning is a branch of artificial intelligence (AI) that allows systems and computers to learn from data and improve their performance without the need for explicit programming for each task. In other words, in machine learning, algorithms and models are developed that are able to identify patterns and relationships in data and make decisions or predictions based on these patterns. In machine learning, many algorithms need to estimate probability distributions. Also, the Gaussian process (GP) is used as a powerful model in machine learning to perform regression tasks [25]. Accordingly, plug-in empirical likelihood (PEL) can be used to more accurately estimate parameters in models that are sensitive to data distribution [25]. Machine learning can be used in a wide variety of applications, including digital social entrepreneurship [26], digital devices and online accounts [27], digital password design [28], additive manufacturing [29], numerical model design [30], smart car design [31], object detection [32], classification, routing location [33], prediction [34], Baffle-Enhanced Scour Mitigation [35], Financial markets [36], image processing [37,38] and etc.

Deep learning can be considered an important branch of machine learning. CNN and recurrent neural networks are important deep-learning architectures [39]. Convolutional graph networks were first introduced in 2016 by Michael DeFerrard and colleagues [40]. These scientists were the first to apply graph spectral theory and the idea of signal processing to graphs. Graph theory has facilitated the derivation of convolution functions within this field. The adjacency and degree matrices play a crucial role when applying graph theory to convolutional networks. An adjacency matrix is used in this theory to create links between each vertex in the graph. The adjacency matrix is incorporated to create the degree matrix. Furthermore, the edges connecting the matrix’s vertices to the diagonal elements in this matrix match exactly. The symbols $D\in R^{N\times N}$ and $W\in R^{N\times N}$ stand for the degree matrix and the graph matrix, respectively.
(2)$$D_{\mathrm{ii}}=\sum_{i}W_{ij}$$In this regard, the Laplacian matrix can be expressed as follows:(3)$$L=D-W\in R^{N\times N}$$
(4)$$L=U\Lambda U^{T}$$

Based on the provided relationships, it can be inferred that the Laplacian matrix is derived by subtracting the degree matrix from the adjacency matrix. The Laplacian matrix utilizes singular value decomposition to calculate the basic functions of the graph, which are then used to determine the corresponding matrix. Moreover, the Laplacian matrix can be defined by establishing a connection between it and the matrices of eigenvectors and singular values. The eigenvectors of the Laplacian matrix are represented by the columns of the eigenvector matrix. The computation of the Fourier transform can be achieved by utilizing these vectors, and the introduction of the Fourier bases can be accomplished by incorporating the diagonal eigenvalues ($\Lambda =diag([\lambda_{0,\ldots ,}\lambda_{N-1}])$) into the following relationship:(5)$$U=[u_{0},\ldots ,u_{N-1}]\in R^{N\times N}$$

Simply explained, the Fourier transform and inverse Fourier transform of an arbitrary signal are defined as follows:(6)$$\hat{q}=U^{T}q$$
(7)$$q=UU^{T}q=U\hat{q}$$

The above relations depict the Fourier transform of the graph ($\hat{q}$) as well as the feature vector for a signal such as q with Fourier bases and the Fourier transform of the graph. The deviation of the graph can be computed using the Fourier transform of two signals in the graph’s domain. For example, the convolution of two signals (x and z) with the associated operator (${*}_{g}$) is depicted below:(8)$$x{*}_{g}=U((U^{T}x)⊙(U^{T}z))$$
where ⊙ symbolizes the element-wise Hadamard product and is calculated between the graph Fourier transformed signals. The filter function describes how to use the graph convolution operator with neural networks. The following x-filtered relation g(L) is displayed:(9)$$x{*}_{g}=U((U^{T}x)⊙(U^{T}z))$$

The Laplacian matrix and its decomposition into singular values and eigenvectors can be used to describe the graph convolution in the following way [40]:(10)$$\begin{array}{l} y=g(L)z=Ug(\Lambda )U^{T}z\\ =U(g(\Lambda ))⊙(U^{T}z)\\ =U(U^{T}(U_{g}(\Lambda )))⊙(U^{T}z)\\ =z{*}_{g}(Ug(\Lambda )) \end{array}$$

## 3. Materials and Methods

The proposed model for the automatic recognition of emotions will be considered in this section. Figure 2 shows the main chart related to the proposed method. This section includes the recording of EEG signals, the developed architecture, and how to divide the data.

### 3.1. Data Gathering

This section describes the process of obtaining a database for recognizing emotions from EEG signals. The database collection involved three types of emotions: positive, negative, and neutral. EEG data were collected from 20 master’s students (10 women and 10 men) ranging in age from 17 to 32 years at Tabriz University’s signal processing laboratory. All participants in the medical test had a BMI of 19 to 24, and no previous illnesses were recorded on all records in the morning, ensuring that the participants did not become tired during the test. To assess positive and negative emotions, a paper version of the 9-grade SAM evaluator test was used during the testing phase. A score below 3 was considered low, while a score above 6 was considered high. During the signal recording process, all participants in the experiment provided informed consent. Participants had the option to leave the experiment at any time if they did not wish to continue. Additionally, all participants had no history of illness and were required to abstain from consuming medication, alcoholic and energy drinks, and caffeinated beverages for 72 h prior to the test. In addition, participants were told not to use hair conditioner and to take a bath prior to the test. Finally, before the start of the signal recording, each subject answered a questionnaire about depressive mood. One of the participants in the experiment is seen in Figure 3.

The 21-channel Encephalan device by the Medicom Company of Russia was used to record the signal. In terms of electrode arrangement on the head, the international standard system 10–20 was used. The sampling frequency during recording was 500 Hz, with a matching impedance of 10 kilohms. The laboratory signal recording device had 21 electrodes, 19 of which were available due to the reference placement of A1 and A2 channels. A special cap was used to ensure the comfort and proper placement of the electrodes on the head.

Music has been used to evoke positive and negative emotions in people. Ten happy and sad songs were chosen for each participant. Each song was played for 1 min to induce positive and negative emotions in each participant. Out of the total duration of 1 min allocated for each participant, only 20 s of each song were exclusively utilized for processing. A 20-s window of each song has been selected from the middle of the recorded EEG signal in order to maximize emotional arousal. For each song, in addition, a 10-s period of silence was included between each piece to create a neutral state. Headphones are used to play music to minimize noise for the EEG signal recording device. The choice of music to evoke positive and negative emotions is determined by [41]. Table 1 shows specific information about the music selected for each emotion. In addition, Figure 4 shows how the music was performed for the participants.

### 3.2. Pre-Processing

The first step in pre-processing is to select the electrodes. As previously stated, increasing the number of electrodes for signal recording improves the accuracy and quality of EEG signals. However, increasing the number of electrodes increases computational efficiency, making the proposed model unsuitable for real-time applications. According to [14,15,16], only three C_3_, C_4_, and P_Z_ electrodes were used for signal processing in this study, and the remaining channels were excluded.

According to Figure 4, in order to remove the overlapping with the neutral phase, the middle of the one-minute recording of the EEG signal related to positive or negative emotions has been used. A 20-s window has been used for this part of the pre-processing stage.

The second step is stacking the samples of each class. The third step is augmenting the stacked signal with 50,000 (number of positive emotions of music × 20 s × sampling frequency). The augmentation step has been completed with a GAN. Two sets of 50,000 samples have been generated, and again, the stacking function has been used. For the next pre-processing stage, 125,000 samples of this stacked signal have been considered. The neutral phase of the recordings has 25,000 samples, and four sets of 25,000 samples should be generated with the GAN to have an equal sample of positive and negative emotions. Three sets of 125,000 samples have been generated corresponding to the positive, negative, and neutral classes. It should be emphasized that the use of the GAN is limited to the training set and does not incorporate the test subject data in any evaluation criteria.

The final step is considering a window to generate multiple signals for each class. We have considered generating 1000 sets of each emotion for every participant to impose them on the proposed network. According to 1000, the window size should be 125. We have considered a window with a size equal to 125. In order to overcome overfitting at the training stage, the windowing has been conducted without overlapping. The stride of the window has been considered equal to 125.

The total number of sets for each class considering the number of participants would be equal to 20,000 (20 × 1000). Considering 3 classes of emotion, we have 60,000 sets of signals for the training, validation, and testing of the proposed network.

### 3.3. Graph Design in Our Pipeline

A proximity matrix is produced when the functional connection of the EEG channels has been established. This may be accomplished by evaluating the channels’ correlation and displaying the results as an EEG channel connection matrix. In the sparse approximation of the connectivity matrix, a threshold is established to remove the network adjacency matrix. The suggested model recognizes and classifies features based on the created graph. Figure 5 shows the graph creation graphically.

### 3.4. Architecture

The proposed deep model is used to automatically recognize emotions after constructing the D graph for this subsection. The overall pipeline of the proposed architecture is shown graphically in Figure 6. Based on this figure, after passing through the dropout layer, the recorded EEG signal is transferred to a graph convolutional layer for processing. This layer also includes a Max Pooling and a Leaky-ReLU function. The aforementioned architecture is repeated three times without the dropout layer. On this basis, four layers of graph convolutional are used to extract the dynamic information of EEG signals. Then, each class is scored using a Softmax function (positive, negative, and neutral excitement levels). Figure 7 depicts the aforementioned contents graphically, along with layer details.

According to the figure, the number of graph nodes is equal to 3 channels in every convolutional layer. This way, each vertex in the convolutional layer gets 1000 sets of 125 samples. Table 2 illustrates how to select the coefficients of the Chebyshev polynomial expansion. These coefficients are chosen through trial and error in the proposed architecture.

According to Figure 7, the weight tensors considering the number of sets equal to K are represented in Table 2.

### 3.5. Training, Validation, and Test Series

This research’s architecture is organized utilizing the approach of trial and error. In this way, it can be ensured that the parameter used in the proposed architecture is chosen correctly. The selected optimal parameters are shown in Table 3. In the training phase, 70% of sets of each class have been considered for training corresponding to 42,000 (20 × 700 × 3) sets of each class, 20% for validation corresponding to 12,000 (20 × 200 × 3) sets, and finally 6000 (100 × 20 × 3) sets have been considered for the test phase. The 5-fold cross-validation has been completed for the training and validation sets, and the trained network weights have been obtained. The weight tensor has been considered to calculate the test accuracy of the test data. In addition, we used leave-one-out (LOV) evaluation. With this evaluation, we can ensure that all of the data participate in the training, testing, and validation processes and that the data are not overfitted.

## 4. Experimental Findings

The outcomes of the suggested architecture will be examined in this section. All processes were carried out in Python using a deductive approach [42] and Google Club’s premium version, which includes the Tesla 60 GPU processor. This section contains sub-sections such as the proposed architecture’s optimization results, results for emotion classification, and a comparison with recent research.

### 4.1. Optimization Findings

The results related to the optimization of the proposed architecture are presented in this subsection. Figure 8 illustrates the importance of selecting convolutional graph layers. This figure illustrates that using four convolutional layers is the most efficient and accurate option in terms of computational efficiency and accuracy. Figure 9 also depicts the various Chebyshev polynomial coefficients used in the proposed architecture. As is known, using *X* = 3 has accelerated the architecture’s convergence to the desired value.

### 4.2. Simulation Findings

Figure 10 displays the accuracy and error of the suggested framework for categorizing two and three emotional classes in 200 iterations of the network. According to the figure, it is evident that the network has achieved its stable state after 110 iterations for two-class classification and after 120 iterations for three-class classification. According to the error diagram, it is evident that as the number of repetitions increases, the error for both two-class and three-class classification decreases and eventually reaches its minimum value. Table 4 displays the evaluation results of different indicators, such as accuracy, precision, sensitivity, specificity, and kappa coefficient, for the classification of two and three classes of emotion. It is widely recognized that the classification results for both two and three classes of emotion are consistently above 96%. Figure 11 depicts the receiver operating characteristic (ROC) curve for emotion classification. Based on Figure 11, it is evident that the bisector of the curve on the left side falls within the range of 1 to 0.9 for each emotional class. This aligns with the reference results depicted in the graph. Figure 12 displays the T-SNE diagram pertaining to the initial and final layer of the proposed network for two and three classes of emotion. It is widely recognized that the majority of samples in the final layer are highly distinct from one another, demonstrating the network’s ability to effectively differentiate between classes. Figure 13 depicts the 5-fold cross-validation results, which, as previously stated, the classification results of classes positive, negative, and neutral in various folds are greater than 97%, indicating that the signal was not overfitted.

Table 2 also includes the results from the LOU evaluation. In this type of evaluation, one participant’s data are used as the test data, while the data of other participants are used for training purposes. This procedure is repeated for the number of participants in the experiment. The benefit of this method is that all data will be used to train and test the model. Table 5 shows the results for all repetitions. According to this table, the classification accuracy remains above 90%, and overfitting has not occurred.

### 4.3. Comparison

This subsection will present a comparison between the proposed model and other recent studies. Table 6 displays recent studies, including the methodology employed and the corresponding accuracy achieved. According to the table, the proposed model demonstrates the highest accuracy when compared to recent studies. Specifically, the accuracy of the proposed model is approximately 99% for classifying two classes and 98% for classifying three classes of emotions. In contrast, the accuracy for studies [18,23] is around 98% and 96% respectively. Nevertheless, it would be unjust to make a direct comparison with recent studies because they utilized different databases. Therefore, we have utilized prevalent algorithms employed in recent research to train our recorded data and subsequently compared the data with the findings of recent studies. The algorithms that have been extensively utilized in studies on emotion recognition and classification are CNN, Inception, and U-Net. The results obtained are displayed in Figure 14. The proposed model achieves higher accuracy compared to other models due to its unique and optimal architecture. Furthermore, an additional comparison is provided to substantiate the effectiveness of the deep architecture proposed in this study. To achieve this objective, we utilized widely used algorithms derived from recent research to train our collected data. Subsequently, we compared our findings with the outcomes of recent studies. The algorithms used include feature selection/extraction, manual classification, and feature learning. In order to accomplish this goal, the EEG signal collected was used to calculate standard statistical measures including mean, variance, skewness, kurtosis, peak coefficient, and power. These measures using support vector machine (SVM) [43], multi-layer perceptron (MLP) [44], K-nearest neighbor (KNN) [45] classifiers, the basic CNN [46], and the proposed DFCGN model were classified. In addition, in the second phase, feature learning was performed using SVM, MLP, and KNN algorithms and the proposed model. The obtained results are presented in Table 7. The utilization of the feature-learning method with deep-learning networks has clearly improved the classification accuracy when compared to manual feature extraction. Nevertheless, it is not suitable to manually implement feature extraction in deep-learning networks. In addition, the manual approach is highly efficient in utilizing traditional classification algorithms, such as MLP, SVM, and KNN. However, using manual techniques requires prior expertise in the subject or issue and can improve the computational efficiency of the algorithm.

EEG signals have a low signal-to-noise ratio. This makes it difficult to apply these signals in online applications. For this purpose, the classification algorithm must be able to withstand environmental noise. To assess the proposed model, we artificially added Gaussian white noise [47] to the data and compared the performance of the proposed pipeline to the competing algorithms. Figure 15 shows the results. As can be seen, the proposed model is very resistant to environmental noises, and the classification accuracy decreases as SNR increases. Other algorithms’ classification accuracy has decreased significantly, making their use in real-time applications difficult.

While the suggested approach works effectively, there are still certain restrictions. The lack of emotional lessons is one of the drawbacks. It is feasible to include more emotion classes—such as anger, melancholy, and other emotions—in future studies. Moreover, it would be prudent to increase the experimental database’s size in order to eliminate the need for algorithms like the GAN to enlarge the data. The suggested approach can be applied in real time by using dry electrodes for data recording as well.

## 5. Conclusions

This study focuses on the utilization of graph theory and deep-learning networks to automatically recognize emotions using EEG signals. To achieve this objective, a database was created by exposing 20 participants to musical stimuli that evoked emotions belonging to two or three distinct emotional categories, namely positive, negative, and neutral. Subsequently, the recorded samples are enhanced using GANs before being fed into the proposed model. Subsequently, the process of feature selection/extraction and automatic classification was conducted utilizing four layers of graph convolution. In this study, the classification accuracy for recognizing three (positive, negative, and neutral) and two (positive and negative) classes of emotions was 99% and 98%, respectively, which is very promising when compared to recent research. Moreover, the suggested model has undergone evaluation in noisy settings and has demonstrated highly promising outcomes in comparison to other studies. The suggested model can serve as a main component in BCA systems.

## Figures and Tables

**Figure 1 sensors-24-05883-f001:**
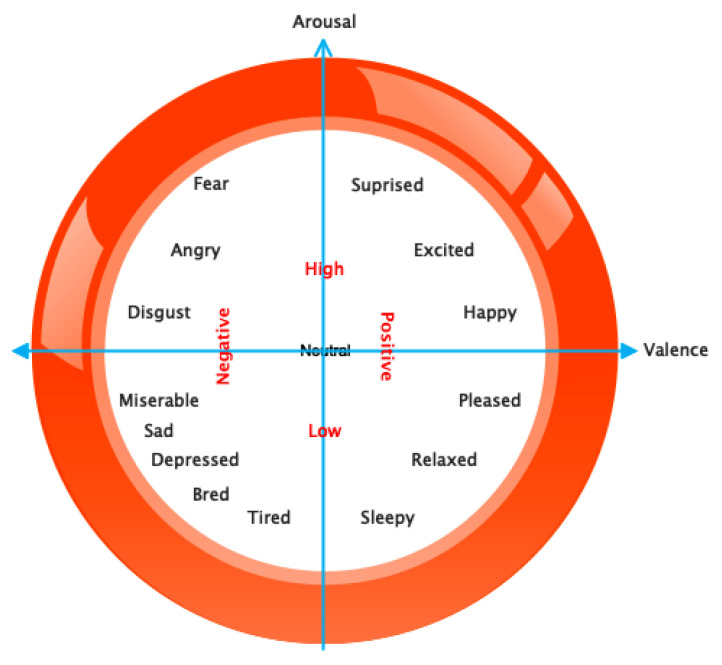
Emotional classes based on Russell’s theory.

**Figure 2 sensors-24-05883-f002:**
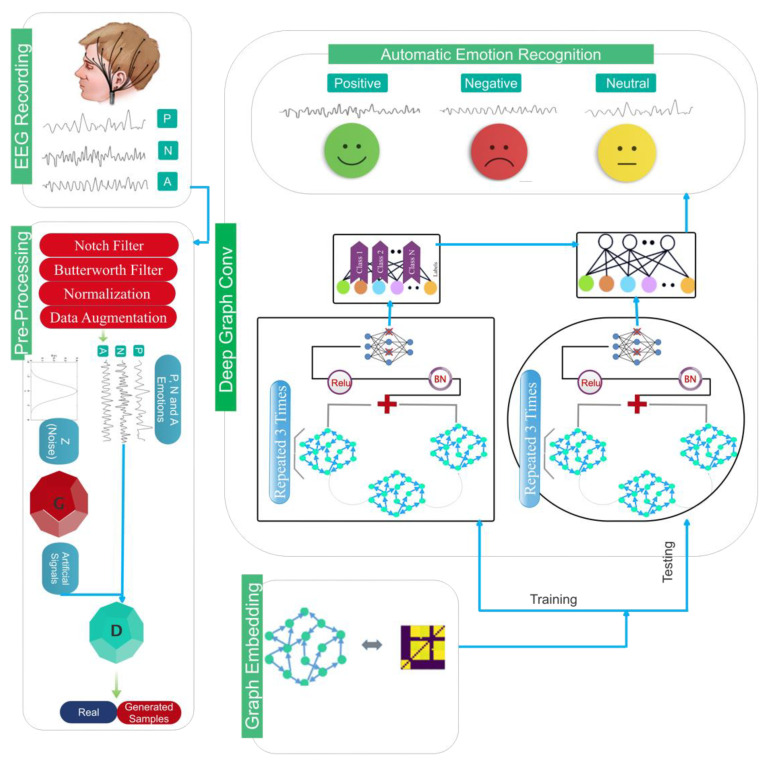
The main pipeline for automatic recognition of emotions using EEG signals.

**Figure 3 sensors-24-05883-f003:**
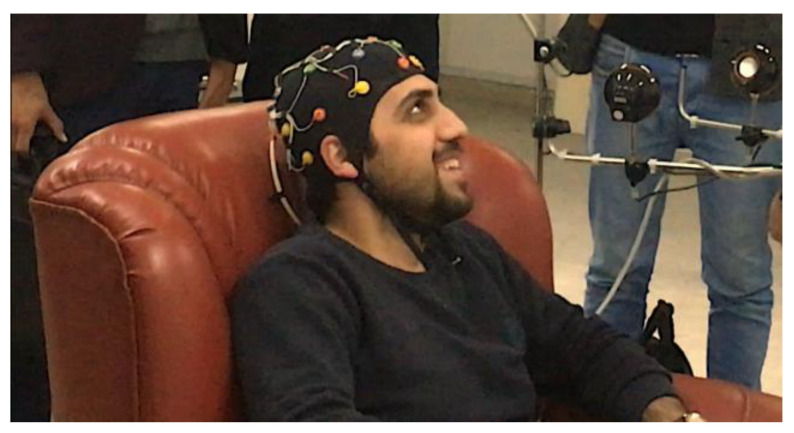
Recording of EEG signals related to one of the participants in the experiment.

**Figure 4 sensors-24-05883-f004:**
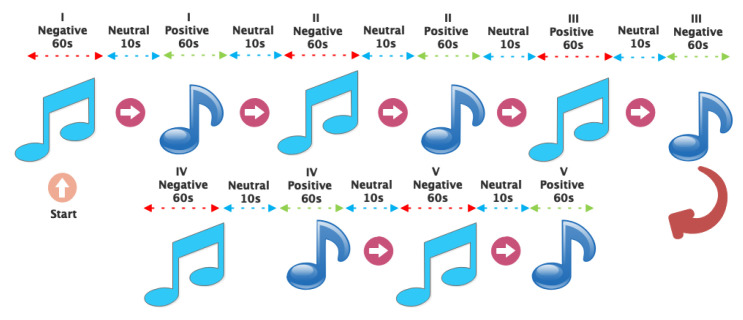
The sequence and timing of the playing of different music to arouse different emotions in the participants.

**Figure 5 sensors-24-05883-f005:**
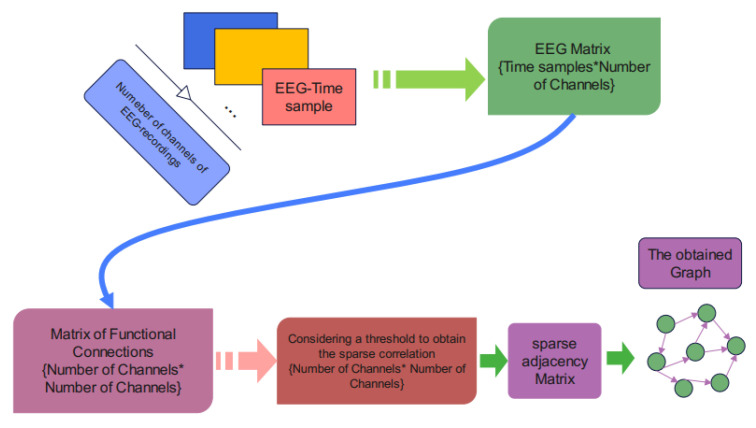
Graph design stage (The sign * in this figure indicates the multiplication sign).

**Figure 6 sensors-24-05883-f006:**
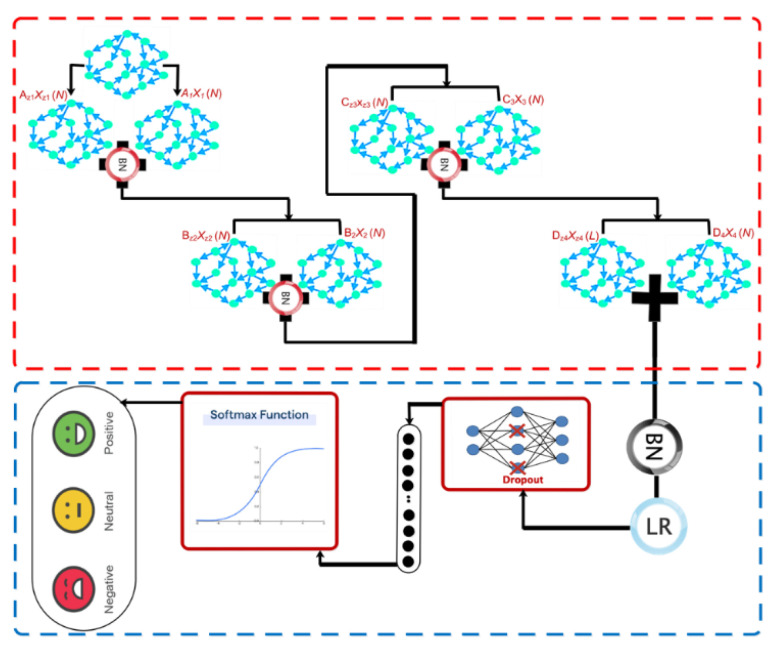
General pipeline for the proposed architecture in this study.

**Figure 7 sensors-24-05883-f007:**
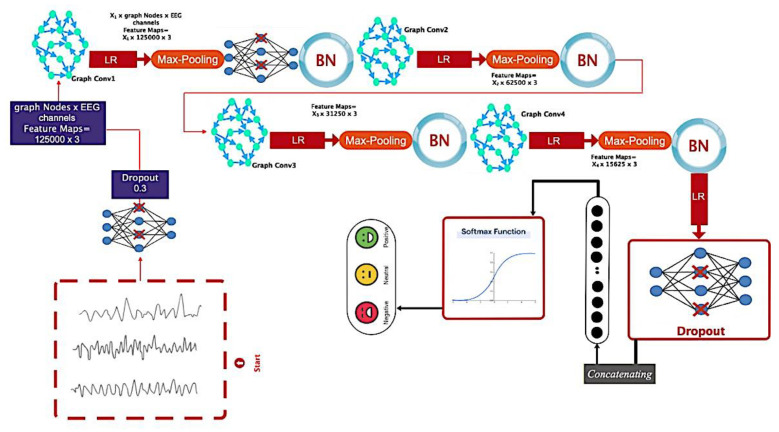
Details related to the pipeline of the proposed architecture.

**Figure 8 sensors-24-05883-f008:**
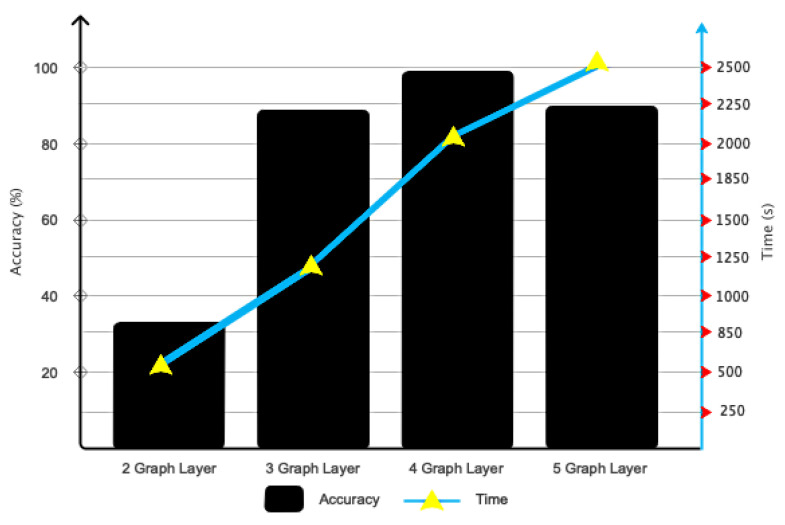
Results related to the use of different numbers of layers in the proposed pipeline.

**Figure 9 sensors-24-05883-f009:**
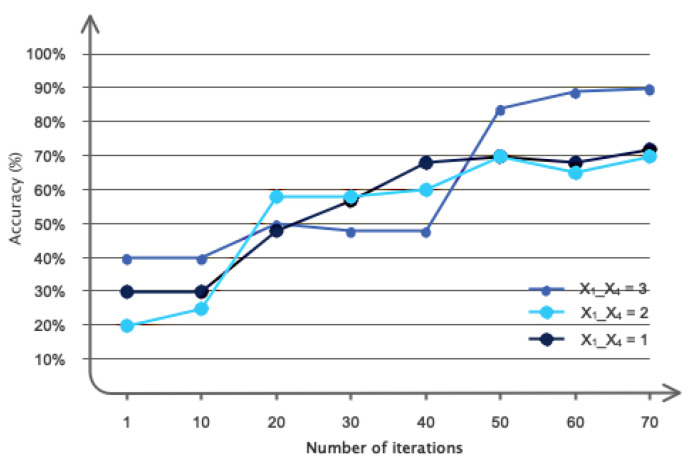
Results related to the use of different polynomial coefficients in the proposed pipeline.

**Figure 10 sensors-24-05883-f010:**
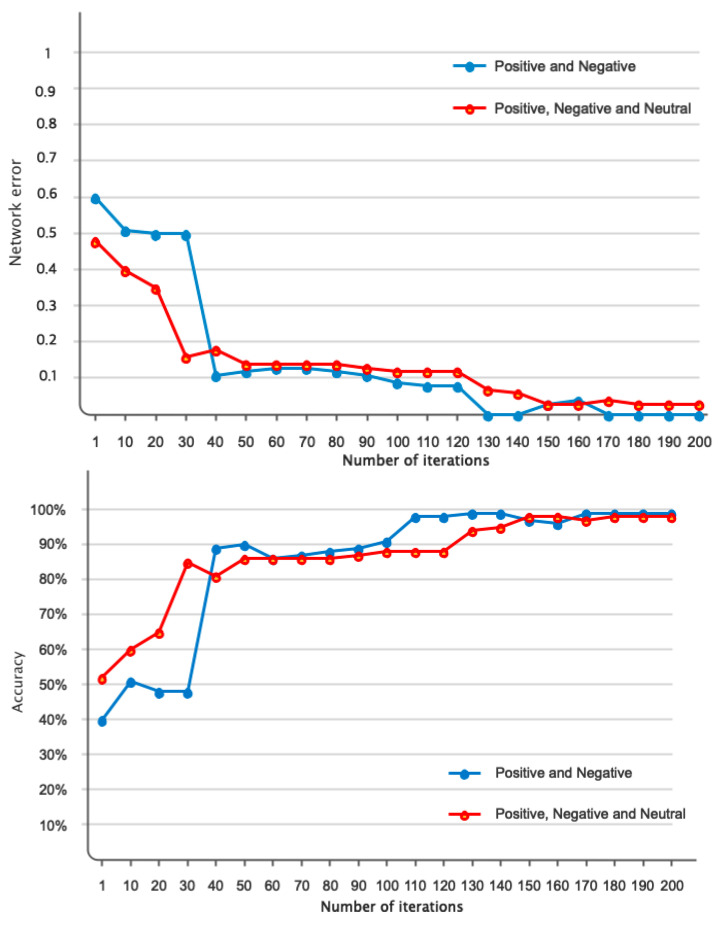
The results related to the correctness and error of the proposed pipeline in 200 repetitions of the network.

**Figure 11 sensors-24-05883-f011:**
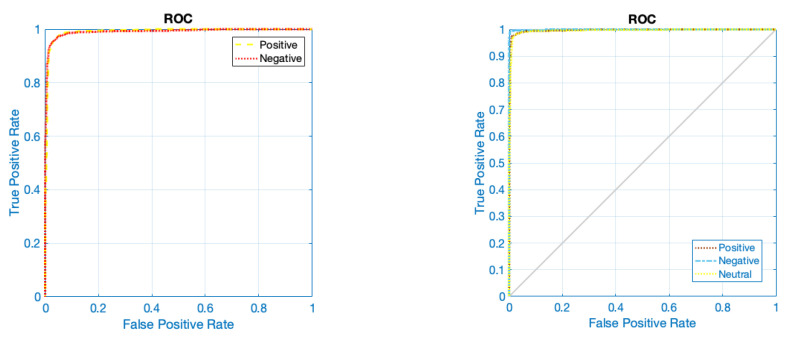
ROC analysis for classification of 2 and 3 classes of emotions.

**Figure 12 sensors-24-05883-f012:**
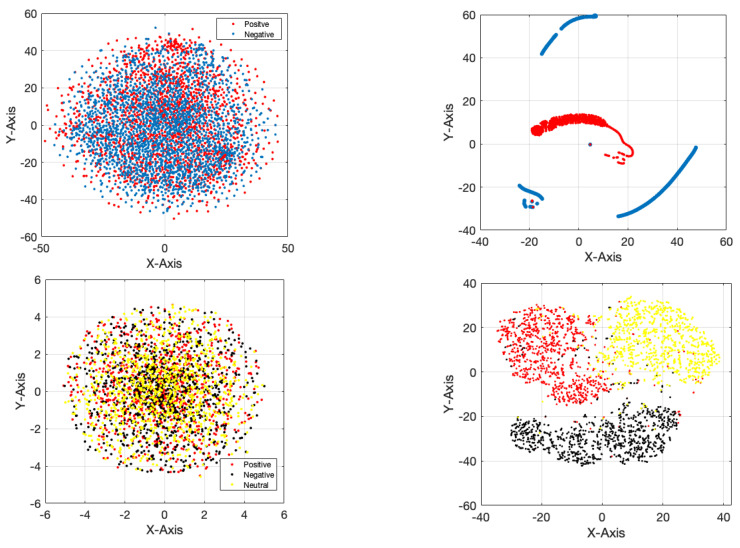
T-SNE chart for the classification of two and three classes of emotions.

**Figure 13 sensors-24-05883-f013:**
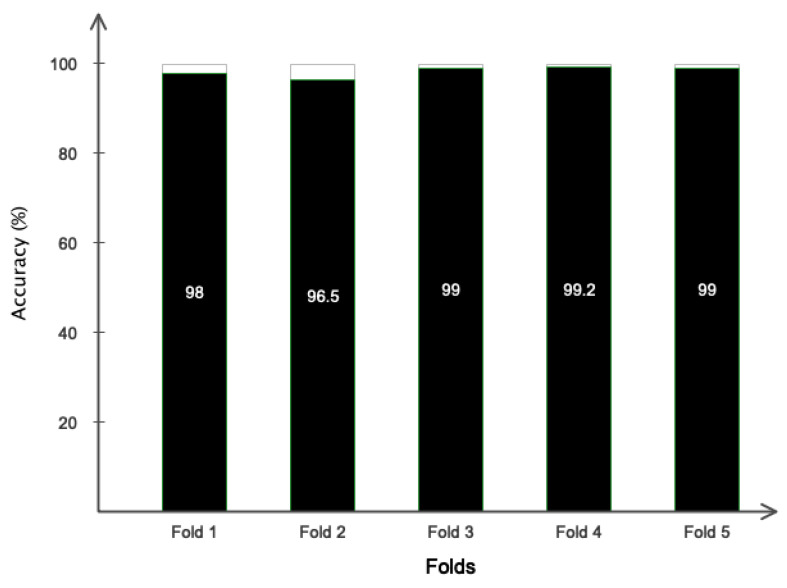
Performance of the proposed model using 5-fold cross-validation.

**Figure 14 sensors-24-05883-f014:**
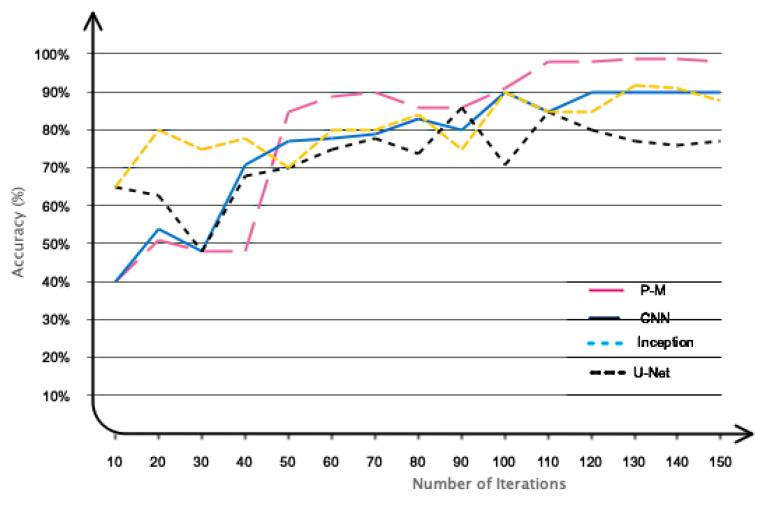
Contrasting the suggested method’s performance with several algorithms.

**Figure 15 sensors-24-05883-f015:**
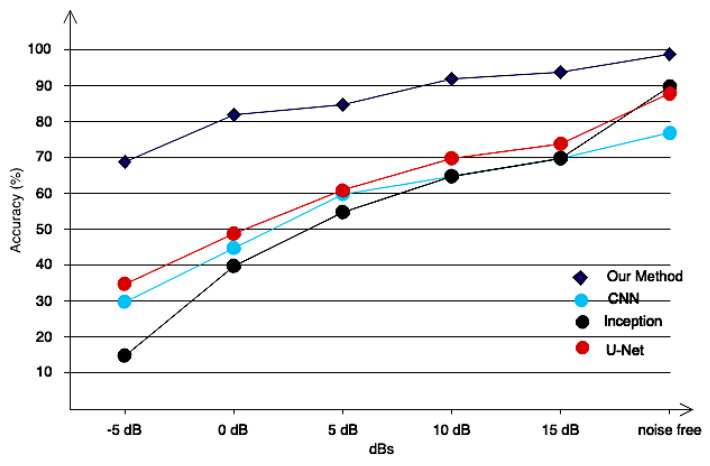
Comparing the resistance of different algorithms against the suggested method in the presence of measurement noise.

**Table 1 sensors-24-05883-t001:** List of music played to arouse emotions.

Emotion	Music Played
**Negative I**	Pishdaramd Esfehani
**Positive I**	Azari 6 and 8
**Negative II**	Pishdaramd Homayoun
**Positive II**	Azari 6 and 8
**Positive III**	Bandari 6 and 8
**Negative III**	Afshari
**Negative IV**	Pishdaramd Esfehani
**Positive IV**	Persion 6 and 8
**Negative V**	Pishdaramad Dashti
**Positive V**	Bandari 6 and 8

**Table 2 sensors-24-05883-t002:** Details related to graph convolutional layers in the proposed architecture.

Layers	Weight Tensor	Bias	Parameters
**GConv I**	(*x*_1_, 125,000/K, 125,000/K)	125,000/K	(15,625,000,000/K × K) × *x*_1_ + (125,000/K)
**GConv II**	(*x*_2_, 125,000/K, 62,500/K)	62,500/K	(7,812,500,000/K × K) × *x*_2_ + (round (62,500/K))
**GConv III**	(*x*_3_, 62,000/K, round (31,250/K))	30,000/K	(1,860,000,000/K × K) × *x*_3_ + (30,000/K)
**GConv IV**	(*x*_4_, round (31,250/K), round (15,625/K))	Round (15,625/K)	(4,588,281,250/K × K) × *x*_4_ + (round (15,625/K))
**Flattening Layer**	(round (31,250/K), 2)	2	(round(62,500/K)) + 2

**Table 3 sensors-24-05883-t003:** Tested parameters in the proposed pipeline.

Parameters	Values	Optimal Value
Batch Size in GAN Optimizer in GAN Number of ConV in GAN Learning Rate in GAN Number of GConv	4, 6, 8, 10, 12 Adam, SGD, Adamax 3, 4, 5, 6 0.1, 0.01, 0.001, 0.0001 2, 3, 4, 5, 6, 7	8 Adamax 6 0.001 4
Batch Size in DFCGN	8, 16, 32	16
Batch normalization	ReLU, Leaky-ReLU, TF-2	Leaky-ReLU
Learning Rate in DFCGN	0.1, 0.01, 0.001, 0.0001, 0.00001	0.001
Dropout Rate	0.1, 0.2, 0.3	0.3
Weight of optimizer	$6\times 10^{-3},4\times 10^{-4},6\times 10^{-5},6\times 10^{-6},6\times 10^{-7}$	$6\times 10^{-6}$
Error function	MSE, Cross Entropy	Cross Entropy
Optimizer in DFCGN	Adam, SGD, Adadelta, Adamax	Adam

**Table 4 sensors-24-05883-t004:** Results of evaluation indices for classification of emotions.

Measurement Index	2-Class	3-Class
**Accuracy**	99.1	98.2
**Sensitivity**	98.4	97.2
**Precision**	99.4	97.8
**Specificity**	97.8	96.3
**Kappa coefficient**	0.8	0.9

**Table 5 sensors-24-05883-t005:** The results related to the evaluation of LOU in order to recognize 3 different emotional classes.

Repetitions	First	Second	Third	Fourth	Fifth	Sixth	Seventh	Eighth	Ninth	Tenth
**ACC (%)**	91	94	92	97	90	93	94	94	95	94
**Repetitions**	Eleventh	Twelfth	Thirteenth	Fourteenth	Fifteenth	Sixteenth	Seventeenth	Eighteenth	Nineteenth	Twentieth
**ACC (%)**	93	97	90	94	92	93	98	91	93	93
**Average Accuracy (%)**								**93.4**		

**Table 6 sensors-24-05883-t006:** The suggested approach is contrasted with more modern approaches.

Research	Datasets	Algorithms	ACC (%)
**Sheykhivand et al.** [14]	Private	CNN + LSTM	97
**Baradaran et al.** [15]	Private	DCNN	98
**Baradaran et al.** [16]	Private	Type 2 Fuzzy + CNN	98
**Yang et al.** [17]	Deap, Seed	SITCN	95
**Hussain et al.** [18]	Deap, Seed	LP-1D-CNN	98.43
**Khubani et al.** [19]	Private	DCNN	97.12
**Peng et al.** [20]	Deap, Seed	Temporal Relative (TR) Encoding	95.58
**Xu et al.** [21]	Private	Functional Connectivity Features	97
**Alotaibi et al.** [22]	Deap, Seed	GoogLeNet DNN	96.95
**Qiao et al.** [23]	Private	CNN-SA-BiLSTM	96.43
**Our Model**	Private	**Graph Theory + CNN**	**99.2 class** **98.3 class**

**Table 7 sensors-24-05883-t007:** Comparing the utilization of an engineering feature extraction method with feature learning in order to discern positive and negative emotions.

Method	Feature Learning (ACC)	Handcrafted Features (ACC)
**KNN**	72%	78%
**SVM**	74%	89%
**CNN**	93%	73%
**MLP**	78%	90%
**P-M**	99%	78%

## Data Availability

The data are private and the University Ethics Committee does not allow public access to the data.

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
