# Peer review of "Automatic Recognition of Multiple Emotional Classes from EEG Signals through the Use of Graph Theory and Convolutional Neural Networks"

_sensors, 2024, doi:10.3390/s24185883_

Round 1
Reviewer 1 Report
Comments and Suggestions for Authors
The authors present a new methodology which combines graph theory and convolutional neural networks for emotion recognition using EEG signals. A database of EEG signals based on musical stimuli was created and a generative adversarial network (GAN) was used for data augmentation. Only three EEG channels (C3, C4 and PZ) were selected for emotion recognition. All signals are preprocessed by using a notch filter and a Butterworth filter. The results indicated that the proposed model has average accuracies of 99% and 98% for two and three classes, respectively. A comparison with other approaches from the state-of-the-art was performed, which showed the effectiveness of the proposed model.
The approach taken by the authors is sound. The paper is well structured and clear.
However, the authors must attend to the following points:
1. Perform the corrections annotated in the PDF file.
2. Improve the quality of all figures.
3. Write tables using text, do not use images as tables.
4. The data augmentation must be only for the training set.
5. Give more details about the database. For instance, total number of samples, number of samples of each class, number of samples in the training, testing and validation datasets. How are testing and validation datasets selected?

The quality of English language is fine. Only minor corrections needed.
Author Response
Original Manuscript ID: “Sensors-3135870”
Original Article Title: “Automatic recognition of multiple emotional classes from EEG signals through the use of graph theory and convolutional neural networks”
To: Guest Editor
Re: Response to reviewers
Dear Editor
Please find below the response to respective reviewers’ comments. We considered all of the comments in detail and did our best to modify the paper in the way they suggested. We believe that the comments have considerably increased the quality of the manuscript. We would be most grateful if you consider the revised manuscript entitled “Automatic recognition of multiple emotional classes from EEG signals through the use of graph theory and convolutional neural networks” for possible publication in the Journal of Bioengineering. We are uploading (a) our point-by-point response to the comments (below) (response to reviewers), (b) an updated manuscript with yellow highlighting indicating changes, and (c) a clean updated manuscript without highlights (PDF main document).
Best regards,
Sebelan Danishvar
Department of Eelectronic and Computer Eng
College of Eng Design and Physical Sciences
Brunel University, UK.
Tel: +44 (0)1895 265089
Fax: +44 (0) 1895 265090
Email: Sebelan.danishvar@brunel.ac.uk
Reviewer#1:
Comments:
The authors present a new methodology which combines graph theory and convolutional neural networks for emotion recognition using EEG signals. A database of EEG signals based on musical stimuli was created and a generative adversarial network (GAN) was used for data augmentation. Only three EEG channels (C3, C4 and PZ) were selected for emotion recognition. All signals are preprocessed by using a notch filter and a Butterworth filter. The results indicated that the proposed model has average accuracies of 99% and 98% for two and three classes, respectively. A comparison with other approaches from the state-of-the-art was performed, which showed the effectiveness of the proposed model. The approach taken by the authors is sound. The paper is well structured and clear.
- While thanking the esteemed reviewer for a thorough review of the manuscript version. We, the authors of the article, believe that your suggestions have been very useful and effective in improving the scientific version of the manuscript. We carefully answered all the questions and suggestions of the esteemed reviewer and added them to the manuscript version.
However, the authors must attend to the following points:
- Perform the corrections annotated in the PDF file.
- The manuscript is revised based on this comment. According to the opinion of the honorable referee, all the forms of the manuscript were qualitatively improved which are highlighted on manuscript.
- Improve the quality of all figures.
- The manuscript is revised based on this comment. According to the reviewer's opinion, all figures of the manuscript were qualitatively improved.
- Write tables using text, do not use images as tables.
- The manuscript is revised based on this comment. According to the reviewer's opinion, all the tables in the article were changed to text.
- The data augmentation must be only for the training set
- As you know, deep learning networks are big data networks and need a lot of data for training. Data augmentation is only used for training data. According to the respected referee's opinion, a 5-fold cross validation was also performed to ensure that the phenomenon of over-fitting did not occur. The findings are as follows:
“Also, to ensure that no overfitting occurred, the data was subjected to 5-fold cross vali-dation. Based on this, we ensure that all data was used during the testing process.
Figure 13 depicts the 5-fold cross validation results, which, as previously stated, the classification results of classes Positive, Negative, and Neutral in various folds are greater than 97%, indicating that the data was not overfitted.
Figure 13. Performance of the proposed model using 5-fold cross validation.
which are highlighted on pages 11 and 15.
- Give more details about the database. For instance, total number of samples, number of samples of each class, number of samples in the training, testing and validation datasets. How are testing and validation datasets selected?
- The manuscript is revised based on this comment. More information about the database has been provided based on the opinion of the respected referee, including the number of samples, the amount of data for each class, and the order of data for the training and evaluation sets:
“The database collection involved three types of emotions: positive, negative, and neutral. EEG data were collected from 20 master's students (10 women and 10 men) ranging in age from 17 to 32 years at Tabriz University's signal processing laboratory. All participants in the medical test had a BMI of 19 to 24, and no previous illnesses were recorded on all records in the morning, ensuring that the participants did not become tired during the test.”
“In data training and evaluation, 70% of the data set was chosen for network training, 20% for network validation, and the remaining 10% was organized for network testing. Also, to ensure that no overfitting occurred, the data was subjected to 5-fold cross validation. Based on this, we ensure that all data was used during the testing process.”
“The data should then be augmented with GAN. The increase in data is due to the unification of classification classes and the prevention of overfitting. Figure 4 shows that the neutral, positive, and negative emotional classes do not have the same sample dimensions. This problem will skew the data toward the majority class in the classification. Furthermore, because deep learning networks are used for feature selection/extraction and classification, the number of samples used to train the network should be maximized. For this purpose, a random vector with a uniform distribution and dimensions of 100 is chosen to generate an output vector with dimensions of for the generating network. This network has six convolutional layers with dimensions of 500, 1000, 1500, 2000, 2500, and 50000. In each convolutional layer, a batch norm and a Leaky-Relu activation function are present. The batch size in layers is 10, and the learning rate is 0.001. The number of repetitions in this network is 150. The discriminator network also checksdimensional signals to see if they are real or not. This network also has six convolutional layers, with a dropout layer in the first layer. This network also uses an ADAM optimizer and cross-entropy cost function. The final layer of the network likewise makes use of the sigmoid activation function. The number of samples rose from 50,000 to 125,000 after employing the GAN network, which can be quite helpful in speeding up the training of the suggested deep network.”
which are highlighted on pages 7, 9, and 11.

Reviewer 2 Report
Comments and Suggestions for Authors
This manuscript introduces an automated method for recognizing multiple emotional states from EEG signals using a combination of graph theory and CNN. The authors have created a database using musical stimuli to elicit distinct emotional responses and have applied GANs to expand the dataset. The results demonstrate high classification accuracy for both binary and ternary emotion categories. The study also assesses the model's performance in the presence of noise, showing its potential for practical applications in BCI systems. The objective of this work is clear, and the proposed technique sounds available. However, several vital issues should be improved, as listed below:
1. This manuscript uses many acronyms and terms. I think it is necessary to add a table regarding their full names and usage. It will be more convenient.
2. This work focuses on CNN. To make the experiments more impressive, I suggest comparing the performance between the conventional machine learning-based method and deep learning-based method under the same conditions so that it will show the advantages of this work.
3. The authors should provide details concerning the overall complexity and time consumption of the proposed networks.
4. In Figure 4, the number of emotional stimuli is different, how to deal with the imbalance data issue in method validation?
5. It is necessary to combine the views of neuroscience and psychology to explain and analyze the results.
6. To facilitate reproducible research, I suggest that the authors release the programming codes of the proposed method and the EEG recordings, and then provide the link in the manuscript. It would have a positive effect on the academic community.
7. What is the tool used for programming in this work? Matlab? Python? Please include the conditions of the experiments, e.g., OS, CPU, GPU, etc.
8. In Table 5, a comprehensive comparative study is required, otherwise, it is difficult to demonstrate the enhancement of the proposed method. The comparison should not only include the accuracy and the methodology, but also contain the data source, time complexity, channels used, computation cost, and so on.
9. Concerning the experiment, is the music heard by the subjects in their native language? It is ensured that this music can evoke the correct emotions without being influenced by individuals.
10. Following the above question, from my understanding, emotional response is a subjective behavior influenced by cultural, experiential, and background factors. Based on that, what is your consideration of individual differences in this work?
Comments on the Quality of English LanguageThe writing still requires further improvement, especially the tense.
Author Response
Reviewer#2:
Comments:
This manuscript introduces an automated method for recognizing multiple emotional states from EEG signals using a combination of graph theory and CNN. The authors have created a database using musical stimuli to elicit distinct emotional responses and have applied GANs to expand the dataset. The results demonstrate high classification accuracy for both binary and ternary emotion categories. The study also assesses the model's performance in the presence of noise, showing its potential for practical applications in BCI systems. The objective of this work is clear, and the proposed technique sounds available. However, several vital issues should be improved, as listed below:
- While thanking the esteemed reviewer for a thorough review of the manuscript version. We, the authors of the article, believe that your suggestions have been very useful and effective in improving the scientific version of the manuscript. We carefully answered all the questions and suggestions of the esteemed reviewer and added them to the manuscript version.
- This manuscript uses many acronyms and terms. I think it is necessary to add a table regarding their full names and usage. It will be more convenient.
The manuscript is revised based on this comment. According to the reviewer's opinion, a table containing abbreviations has been added to the manuscript. which are highlighted on page 18.
Acronym |
phrase |
EEG |
Electroencephalogram |
GANs |
Generative Adversarial Networks |
BCI |
Brain-Computer Interface |
ECG |
Electrocardiogram |
EMG |
Electromyogram |
GSR |
Galvanic Skin Response |
ML |
Machine Learning |
KNN |
K-Nearest Neighbor |
LSTM |
Long-Term Short-Term Memory |
CNN |
Convolutional Neural Networks |
FC |
Fully Connected |
DWT |
Discrete Wavelet Transform |
TR |
Temporal Relative |
STFT |
Short-Time Fourier Transform |
G |
Generator |
D |
Discriminator |
ROC |
Receiver Operating Characteristic |
- This work focuses on CNN. To make the experiments more impressive, I suggest comparing the performance between the conventional machine learning-based method and deep learning-based method under the same conditions so that it will show the advantages of this work.
- The manuscript is revised based on this comment. Yes, the honorable judge's opinion is absolutely correct. For this purpose, manual machine learning methods were compared to the proposed deep model. The results are as follows:
“Furthermore, an additional comparison is provided to substantiate the effectiveness of the deep architecture proposed in this study. To achieve this objective, we utilized widely used algorithms derived from recent research to train our collected data. Subsequently, we compared our findings with the outcomes of recent studies. The algorithms used include feature selection/extraction, manual classification, and feature learning. In order to accomplish this goal, the data collected was used to calculate standard statistical measures including mean, variance, skewness, kurtosis, peak coefficient, and power. These measures using Support Vector Machine (SVM), Multi-Layer Perceptron (MLP), K-Nearest Neighbor (KNN) classifiers, the basic CNN and the proposed DFCGN model were classified. In addition, in the second phase, feature learning was performed using SVM, MLP, KNN algorithms, and the proposed model. The obtained results are presented in Table 6. The utilization of feature learning method with deep learning networks has clearly improved the classification accuracy when compared to manual feature extraction. Nevertheless, it is not suitable to manually implement feature ex-traction in deep learning networks. In addition, the manual approach is highly efficient in utilizing traditional classification algorithms, such as MLP, SVM, and KNN. However, using manual techniques requires prior expertise in the subject or issue and can improve the computational efficiency of the algorithm.”
Table 6. Comparing the utilization of an engineering feature extraction method with feature learning in order to discern positive and negative emotions.
Method |
Feature learning (ACC) |
Hand crafted features (ACC) |
|
KNN |
72% |
78% |
|
SVM |
74% |
89% |
|
CNN |
93% |
73% |
|
MLP |
78% |
90% |
|
P-M |
99% |
78% |
|
which are highlighted on page 16.
- The authors should provide details concerning the overall complexity and time consumption of the proposed networks.
- The opinion of the honorable judge is absolutely correct. However, the computational complexity of the algorithm is presented in section 4.1 and its results are as follows:
“Figure 7 illustrates the importance of selecting convolutional graph layers. This figure illustrates that using four convolutional layers is the most efficient and accurate option in terms of computational efficiency and accuracy.”
Figure 7. Results related to the use of different number of layers in the proposed pipeline.
which are highlighted on page 11.
- In Figure 4, the number of emotional stimuli is different, how to deal with the imbalance data issue in method validation?
- Yes, the honorable judge's opinion is absolutely correct. However, this is mentioned in the manuscript. To avoid data bias towards the majority class, the data was artificially increased using generative adversarial networks, ensuring that the amount of data in each class is equal.
“The data should then be augmented with GAN. The increase in data is due to the unification of classification classes and the prevention of overfitting. Figure 4 shows that the neutral, positive, and negative emotional classes do not have the same sample dimensions. This problem will skew the data toward the majority class in the classification. Furthermore, because deep learning networks are used for feature selection/extraction and classification, the number of samples used to train the network should be maximized. For this purpose, a random vector with a uniform distribution and dimensions of 100 is chosen to generate an output vector with dimensions of for the generating network. This network has six convolutional layers with dimensions of 500, 1000, 1500, 2000, 2500, and 50000. In each convolutional layer, a batch norm and a Leaky-Relu activation function are present. The batch size in layers is 10, and the learning rate is 0.001. The number of repetitions in this network is 150. The discriminator network also checksdimensional signals to see if they are real or not. This network also has six convolutional layers, with a dropout layer in the first layer. This network also uses an ADAM optimizer and cross-entropy cost function. The final layer of the network likewise makes use of the sigmoid activation function. The number of samples rose from 50,000 to 125,000 after employing the GAN network, which can be quite helpful in speeding up the training of the suggested deep network.”
- It is necessary to combine the views of neuroscience and psychology to explain and analyze the results.
- The manuscript is revised based on this comment. Based on the opinion of the respected referee, we have presented the relationship between neuroscience and neuroscience to detect emotions as follows:
“Understanding the relationship between EEG components and different emotions is a crucial scientific inquiry in cognitive neuroscience. It is essential for developing accurate recognition models. According to a report, there is a positive correlation between joyful music and the energy in the theta band (4∼7 Hz) near the midline of the prefrontal cortex. Furthermore, previous studies have found a correlation between the valence and arousal of musical stimuli and the asymmetry of frontal alpha (8∼13 Hz) activity. When subjects are exposed to happy music, the EEG activity in the left frontal region of the brain is more pronounced compared to the activity in the right frontal region. Conversely, playing sad music yields the opposite outcome. However, certain studies did not observe any effects in the alpha frequency range, but instead discovered a correlation in the beta-2 frequency range (18∼22 Hz). The impact of high-frequency EEG, specifically the Beta-2 band (18∼22 Hz), Beta-3 band (22∼30 Hz), and Gamma band (30∼45 Hz), on emotions has also been confirmed. The sensation of happiness leads to a reduction in beta-2 power in the frontal central areas, as well as a decrease in beta-3 and gamma power across the entire cortex. When experiencing anger, there is an increase in power in the Beta-2, Beta-3, and Gamma bands in the frontal regions of both hemispheres.”
which are highlighted on page 3.
- To facilitate reproducible research, I suggest that the authors release the programming codes of the proposed method and the EEG recordings, and then provide the link in the manuscript. It would have a positive effect on the academic community.
- The honorable judge's opinion is absolutely correct. Unfortunately, the university's ethics committee prohibits the publication of recorded codes and data for free access.
- What is the tool used for programming in this work? Matlab? Python? Please include the conditions of the experiments, e.g., OS, CPU, GPU, etc.
- The manuscript is revised based on this comment. All processes were carried out in Python using Google Club's premium version, which includes the Tesla 60 GPU processor.
which are highlighted on page 11.
- In Table 5, a comprehensive comparative study is required, otherwise, it is difficult to demonstrate the enhancement of the proposed method. The comparison should not only include the accuracy and the methodology, but also contain the data source, time complexity, channels used, computation cost, and so on.
- Yes, the opinion of the respected referee is completely correct, we have added the data used in previous studies to Table 5. However, it is not possible to present the computational complexity of each study in this Table; Because most of the computational complexity studies have not presented their proposed method. In addition, the computational complexity of the current research is presented in Figure 3.
which are highlighted on page 16.
- Concerning the experiment, is the music heard by the subjects in their native language? It is ensured that this music can evoke the correct emotions without being influenced by individuals.
- Yes, all the music, as described in Table 3, was played for the participants in the native language of the people, which is Persian.
- Following the above question, from my understanding, emotional response is a subjective behavior influenced by cultural, experiential, and background factors. Based on that, what is your consideration of individual differences in this work?
- According to the referee's opinion, it is widely recognized that there are fundamental and primary emotions that are universally understood, including happiness, fear, anger, disgust, sadness, and surprise. However, neuroscientists and researchers lack agreement regarding the essence of emotions. There are two perspectives on emotions: one approach views emotions as overall states of individuals, while the other perspective regards emotions as physiological interactions.1 Envision an individual operating a motor vehicle when another vehicle approaches and forces them to veer off the road. Initially, the individual likely undergoes feelings of fear and anger. According to one perspective, fear arises from the belief that one may be experiencing anger, which is caused by the driver who has recently endangered them. Thagard [1], Oately [2], and Nussbaum [3] support the initial approach. Oately exemplified the profound connection between genuine emotions and the achievement of objectives. Essentially, individuals experience happiness as they make progress towards their objectives, and sadness when they are unsuccessful. Individuals experience fear when they encounter difficulties or perceive a sense of danger. Thus, emotions can be regarded as a comprehensive depiction of our difficulties.1 Unlike the first perspective, the second approach places greater importance on the physical and physiological interactions. When someone forces a car driver to veer off the road, their heart rate, blood pressure, and respiration rate elevate. Emotions such as fear or anger are generated by the brain's reactions to physiological changes, rather than being influenced by the interpretation of the situation.”
[1] Thagard P. Mind: Introduction to Cognitive Science. Cambridge, MA: MIT press; 2005.
[2] Oately K. Best Laid Schemes: The Psychology of Emotions. Cambridge: Cambridge University Press; 1992.
[3] Nussbaum MC. Upheavals of Thought. Cambridge: Cambridge University Press; 2001.

Reviewer 3 Report
Comments and Suggestions for Authors
Reviewer’s Report on the manuscript entitled:
Automatic recognition of multiple emotional classes from EEG signals through the use of graph theory and convolutional neural networks
The authors presented an automatic method using electroencephalogram (EEG) signals and based on combining graph theory with convolutional networks for emotion recognition. They compared their method with some other methods and achieved good accuracy. From the methodological point of view, the applied methods are known, just the application is new to this specific dataset. The results are interesting but needs improvement. The literature review, figure quality, and presentation should also be improved. Please see below my comments.
Major comments:
Literature review on artifacts corrupting EEG signals:
Lines 58-81. Please review and discuss the following article that proposes a novel baseline removal from EEG signals for emotion classification. In the discussion section, please also elaborate on how you can improve the accuracy of your model by a more robust artifact and baseline noise removal in the light of this article:
“A Novel Baseline Removal Paradigm for Subject-Independent Features in Emotion Classification Using EEG” (Bioengineering, 2023)
Line 86. The EEG signals are also contaminated by eyeblink and muscle artifacts as well as baseline noise. Please see and include the article above and the following article:
"Review of challenges associated with the EEG artifact removal methods” (2021)
There is no discussion on how you handled the artifacts in EEG signals caused by eyeblink or muscle movement. Please elaborate. How much does the artifacts bias your results?
Line 397. There are publicly available datasets such as DEAP and SEED that you can use for the sake of comparison. Please see the articles above.
I like to see some EEG signals (their channels) and some discussion on different spectral bands in EEG for emotion classification (i.e., alpha, beta, gamma, etc. which one is more important?)
The conclusion section is short. Please expand by including some quantitative results, limitations and future direction.
Minor comments:
Line 19. “is incorrect” is not the right expression here. “limited” is a better word. Please rephrase the whole sentence.
Lines 99-153. This is a very large paragraph. Please break this sentence into multiple paragraphs.
Eq. (8): please define the product symbol.
Line 34. Please avoid using “We”, “Our”, “us”. I suggest using the third person.
Figure 2. The quality of this figure should be improved, and the font size should be increased.
Line 234. Section 3 should be called Materials and Methods.
Fig. 3. Did you obtain formal consent of the subject for displaying their data and photos?
The font size of Figures should be increased. This includes Figs. 2 and 5.
Figure 10. It is not clear whether the rows are for truth or predicted. The confusion matrix should clearly mention what row/column is truth/predicted.
Line 429.The word “data” is plural. Please check and correct this grammar issue everywhere.
Thank you!
Regards,
Comments on the Quality of English LanguageThere are many typos/grammar/punctuation issues that should be checked and corrected. I listed some below:
Line 19. “is incorrect” is not the right expression here. “limited” is a better word. Please rephrase the whole sentence.
Lines 99-153. This is a very large paragraph. Please break this sentence into multiple paragraphs.
Eq. (8): please define the product symbol.
Line 34. Please avoid using “We”, “Our”, “us”. I suggest using the third person.
Line 234. Section 3 should be called Materials and Methods.
Line 429.The word “data” is plural. Please check and correct this grammar issue everywhere.
Author Response
Reviewer#3:
Comments:
This paper looks interesting, but some points need to be revised:
The authors presented an automatic method using electroencephalogram (EEG) signals and based on combining graph theory with convolutional networks for emotion recognition. They compared their method with some other methods and achieved good accuracy. From the methodological point of view, the applied methods are known, just the application is new to this specific dataset. The results are interesting but needs improvement. The literature review, figure quality, and presentation should also be improved. Please see below my comments.
- While thanking the esteemed reviewer for a thorough review of the manuscript version. We, the authors of the article, believe that your suggestions have been very useful and effective in improving the scientific version of the manuscript. We carefully answered all the questions and suggestions of the esteemed reviewer and added them to the manuscript version.
Literature review on artifacts corrupting EEG signals:
- Lines 58-81. Please review and discuss the following article that proposes a novel baseline removal from EEG signals for emotion classification. In the discussion section, please also elaborate on how you can improve the accuracy of your model by a more robust artifact and baseline noise removal in the light of this article:
“A Novel Baseline Removal Paradigm for Subject-Independent Features in Emotion Classification Using EEG” (Bioengineering, 2023)
- The manuscript is revised based on this comment. Thanks to the reviewer's opinion, we reviewed the requested study in the study section of previous studies as follows and added it to the manuscript:
“Several methods have been proposed for removing EEG signal artifacts. Some of these methods include repeating experiments to increase SNR [12]. Others use new methods, such as InvBase, to remove the baseline power prior to extracting the features [13].”
which are highlighted on page 3 and refs [12, 13].
- Line 86. The EEG signals are also contaminated by eyeblink and muscle artifacts as well as baseline noise. Please see and include the article above and the following article:
"Review of challenges associated with the EEG artifact removal methods” (2021)
- Thanks to the reviewer's opinion, we reviewed the requested study in the study section of previous studies as follows and added it to the manuscript:
“Several methods have been proposed for removing EEG signal artifacts. Some of these methods include repeating experiments to increase SNR [12]. Others use new methods, such as InvBase, to remove the baseline power prior to extracting the features [13].”
which are highlighted on page 3 and refs [12, 13].
- There is no discussion on how you handled the artifacts in EEG signals caused by eyeblink or muscle movement. Please elaborate. How much does the artifacts bias your results?
- Yes, the honorable judge's opinion is absolutely correct. However, we asked the participants to close their eyes while we recorded EEG signals. This is done to prevent noise from being caused by blinking artifacts on the EEG signal, eliminating the need to use different filters to remove these artifacts.
- Line 397. There are publicly available datasets such as DEAP and SEED that you can use for the sake of comparison. Please see the articles above.
- Yes, the honorable judge's opinion is absolutely correct. However, these databases are organized around non-musical stimuli. For this reason, it is illogical to compare the current database, which is based on musical stimulation, to other databases such as DEAP and SEED. Because the type of musical stimulus affects the participants' emotional arousal.
- I like to see some EEG signals (their channels) and some discussion on different spectral bands in EEG for emotion classification (i.e., alpha, beta, gamma, etc. which one is more important?
- The focus of this study is on feature learning method and feature extraction from EEG signal is not in the scope of this research. However, with respect to the opinion of the respected referee, we have examined the different rhythms of the EEG signal in positive and negative emotions as follows:
“it is widely recognized that there are fundamental and primary emotions that are universally understood, including happiness, fear, anger, disgust, sadness, and surprise. However, neuroscientists and researchers lack agreement regarding the essence of emotions. There are two perspectives on emotions: one approach views emotions as overall states of individuals, while the other perspective regards emotions as physiological interactions.1 Envision an individual operating a motor vehicle when another vehicle approaches and forces them to veer off the road. Initially, the individual likely undergoes feelings of fear and anger. According to one perspective, fear arises from the belief that one may be experiencing anger, which is caused by the driver who has recently endangered them. Thagard [1], Oately [2], and Nussbaum [3] support the initial approach. Oately exemplified the profound connection between genuine emotions and the achievement of objectives. Essentially, individuals experience happiness as they make progress towards their objectives, and sadness when they are unsuccessful. Individuals experience fear when they encounter difficulties or perceive a sense of danger. Thus, emotions can be regarded as a comprehensive depiction of our difficulties.1 Unlike the first perspective, the second approach places greater importance on the physical and physiological interactions. When someone forces a car driver to veer off the road, their heart rate, blood pressure, and respiration rate elevate. Emotions such as fear or anger are generated by the brain's reactions to physiological changes, rather than being influenced by the interpretation of the situation.”
[1] Thagard P. Mind: Introduction to Cognitive Science. Cambridge, MA: MIT press; 2005.
[2] Oately K. Best Laid Schemes: The Psychology of Emotions. Cambridge: Cambridge University Press; 1992.
[3] Nussbaum MC. Upheavals of Thought. Cambridge: Cambridge University Press; 2001.
- The conclusion section is short. Please expand by including some quantitative results, limitations and future direction.
- Yes, the opinion of the honorable judge is absolutely correct. We have increased the conclusion section and added quantitative results related to the present study as follows:
“This study focuses on the utilization of graph theory and deep learning networks to automatically recognize emotions using EEG signals. To achieve this objective, a database was created by exposing 15 participants to musical stimuli that evoked emotions belonging to two or three distinct emotional categories, namely positive, negative, and neutral. Subsequently, the recorded samples are enhanced using GAN networks before being fed into the proposed model. Subsequently, the process of feature selection/extraction and automatic classification was conducted utilizing four layers of graph convolution. In this study, the classification accuracy for recognizing three (positive, negative and neutral) and two (positive and negative) classes of emotions was 99% and 98%, respectively, which is very promising when compared to recent research. Moreover, the suggested model has undergone evaluation in noisy settings and has demonstrated highly promising outcomes in comparison to other studies. The suggested model can serve as a main component in BCA systems.”
which are highlighted on page 18.
- Minor comments:
- Line 19. “is incorrect” is not the right expression here. “limited” is a better word. Please rephrase the whole sentence.
- The manuscript is revised based on this comment.
- Lines 99-153. This is a very large paragraph. Please break this sentence into multiple paragraphs.
- With respect to the opinion of the honorable judge, the requested item is a matter of taste and it is not possible to separate this paragraph. Because this paragraph examines studies related to previous researches in order to automatically detect emotions.
- (8): please define the product symbol.
- The manuscript is revised based on this comment.
- Line 34. Please avoid using “We”, “Our”, “us”. I suggest using the third person.
- The manuscript is revised based on this comment.
Figure 2. The quality of this figure should be improved, and the font size should be increased.
- Line 234. Section 3 should be called Materials and Methods.
- The manuscript is revised based on this comment.
- 3. Did you obtain formal consent of the subject for displaying their data and photos?
- Yes, written consent has been obtained from the subject in the picture to publish her picture in the article.
- The font size of Figures should be increased. This includes Figs. 2 and 5.
- The manuscript is revised based on this comment.
- Figure 10. It is not clear whether the rows are for truth or predicted. The confusion matrix should clearly mention what row/column is truth/predicted.
- With respect to the opinion of the respected referee, Figure 10 shows the classification results of the samples using the proposed model. The samples related to positive and negative emotions are clearly defined.
- Confusion matrices were modified based on the opinion of the respected referee.
- Line 429.The word “data” is plural. Please check and correct this grammar issue everywhere.
- The manuscript is revised based on this comment.

Round 2
Reviewer 1 Report
Comments and Suggestions for Authors
The authors present a new methodology which combines graph theory and convolutional neural networks for emotion recognition using EEG signals. A database of EEG signals based on musical stimuli was created and a generative adversarial network (GAN) was used for data augmentation. Only three EEG channels (C3, C4 and PZ) were selected for emotion recognition. All signals are preprocessed by using a notch filter and a Butterworth filter. The results indicated that the proposed model has average accuracies of 99% and 98% for two and three classes, respectively. A comparison with other approaches from the state-of-the-art was performed, which showed the effectiveness of the proposed model.
The approach taken by the authors is sound. The paper is well structured and clear.
The authors have successfully attended must of the suggestions. However, they must attend to the following suggestions:
1. Perform corrections annotated in PDF file.
2. Improve the quality of all figures.

Author Response
Reviewer#1:
Comments:
The approach taken by the authors is sound. The paper is well structured and clear.
- While thanking the esteemed reviewer for a thorough review of the manuscript version. We, the authors of the article, believe that your suggestions have been very useful and effective in improving the scientific version of the manuscript. We carefully answered all the questions and suggestions of the esteemed reviewer and added them to the manuscript version.
The authors have successfully attended must of the suggestions. However, they must attend to the following suggestions:
- Perform corrections annotated in PDF file.
- The manuscript is revised based on this comment. According to the opinion of the honorable referee, all the forms of the manuscript were qualitatively improved which are highlighted on manuscript.
- Improve the quality of all figures.
- The manuscript is revised based on this comment. According to the reviewer's opinion, all figures of the manuscript were qualitatively improved.
Reviewer 2 Report
Comments and Suggestions for Authors
The authors have completed the revisions and answered my questions. Now, the technical content of this work is clearly written and justified. It can be accepted.
Author Response
Reviewer#2:
Comments:
Thank you for addressing my comments satisfactorily and improving your manuscript. I have some minor editorial suggestions that can be done during proofreading (if accepted by editors):
- While thanking the esteemed reviewer for a thorough review of the manuscript version. We, the authors of the article, believe that your suggestions have been very useful and effective in improving the scientific version of the manuscript. We carefully answered all the questions and suggestions of the esteemed reviewer and added them to the manuscript version.
Line 66. The figure number is incorrect. It is Figure 1.
The manuscript is revised based on this comment.
Line 401. This is Figure 13 not Figure 1. Please carefully check the order of Figures and references to them in the text.
The manuscript is revised based on this comment.
Grammar issue: The word "data" is plural. There are several places that you treated "data" as single, e.g., lines 305, 310, 338, 359, 360, 404, 436, etc.
The manuscript is revised based on this comment.
Reviewer 3 Report
Comments and Suggestions for Authors
Dear authors,
Thank you for addressing my comments satisfactorily and improving your manuscript. I have some minor editorial suggestions that can be done during proofreading (if accepted by editors):
Line 66. The figure number is incorrect. It is Figure 1.
Line 401. This is Figure 13 not Figure 1. Please carefully check the order of Figures and references to them in the text.
Grammar issue: The word "data" is plural. There are several places that you treated "data" as single, e.g., lines 305, 310, 338, 359, 360, 404, 436, etc.
Regards,
Comments on the Quality of English LanguageThere are some grammar/style issues that need to be checked and corrected. For example, "data" is plural. Also Figure numbers and references to them should be checked.
Author Response

(The authors gave the same response as above.)
